# Shrinking Bouma's window: How to model crowding in dense displays

**Alban Bornet**[1]⊙*, **Adrien Doerig**[1,2]⊙, **Michael H. Herzog**[1], **Gregory Francis**[3], **Erik Van der Burg**[4,5]

**1** Laboratory of Psychophysics, Brain Mind Institute, Ecole Polytechnique Fédérale de Lausanne (EPFL), Lausanne, Switzerland, **2** Donders Institute for Brain, Cognition and Behaviour, Radboud University, Nijmegen, The Netherlands, **3** Department of Psychological Sciences, Purdue University, West Lafayette, Indiana, United States of America, **4** TNO, Human Factors, Soesterberg, The Netherlands, **5** Brain and Cognition, University of Amsterdam, Amsterdam, The Netherlands

⊙ These authors contributed equally to this work.
* alban.bornet@epfl.ch

**Data Availability Statement:** The code to reproduce all models' results and the data of the human experiment are available at https://bitbucket.org/albornet/shrinking_boumas_window.

## Abstract

In crowding, perception of a target deteriorates in the presence of nearby flankers. Traditionally, it is thought that visual crowding obeys Bouma's law, i.e., all elements within a certain distance interfere with the target, and that adding more elements always leads to stronger crowding. Crowding is predominantly studied using sparse displays (a target surrounded by a few flankers). However, many studies have shown that this approach leads to wrong conclusions about human vision. Van der Burg and colleagues proposed a paradigm to measure crowding in dense displays using genetic algorithms. Displays were selected and combined over several generations to maximize human performance. In contrast to Bouma's law, only the target's nearest neighbours affected performance. Here, we tested various models to explain these results. We used the same genetic algorithm, but instead of selecting displays based on human performance we selected displays based on the model's outputs. We found that all models based on the traditional feedforward pooling framework of vision were unable to reproduce human behaviour. In contrast, all models involving a dedicated grouping stage explained the results successfully. We show how traditional models can be improved by adding a grouping stage.

## Author summary

To understand human vision, psychophysical research usually focuses on simple stimuli. Vision is often described as a cascade of feed-forward computations in which local feature detectors pool information along the processing hierarchy to form complex and abstract features. Crowding can be modelled within this framework by the pooling of information from one processing stage to the next. This naturally explains Bouma's law, a hallmark of crowding according to which only elements within a certain region, often proposed to be half the target eccentricity, interfere with the target. However, pooling models are strongly challenged by recent experimental results, because Bouma's law does not hold for more complex stimuli. Visual elements far beyond Bouma's window can increase or alleviate

**Funding:** AB was supported by the European Union's Horizon 2020 Framework Programme for Research and Innovation under the Specific Grant Agreements No. 785907 (Human Brain Project SGA2) and No. 945539 (Human Brain Project SGA3). AD was supported by the Swiss National Science Foundation grants n.176153 "Basics of visual processing: from elements to figures" and n.191718 "Towards machines that see like us: human eye movements for robust deep recurrent neural networks". The funders had no role in study design, data collection and analysis, decision to publish, or preparation of the manuscript.

**Competing interests:** The authors have declared that no competing interests exist.

crowding. In addition, Van der Burg and colleagues showed that only the nearest neighbours interfere with the target in dense displays. Hence, Bouma's window can shrink too. Here, we aimed at modelling the range of crowding in dense displays. From previous studies, we know that visual crowding cannot be explained without grouping and segmentation. We compared the performance of different models of vision to the human data of Van der Burg and colleagues. We found that all models based on the traditional pooling framework of vision failed to reproduce the human data, whereas all models that included grouping and segmentation processes were successful in this respect. We concluded that grouping and segmentation processes naturally and consistently explain the difference between simple and complex displays in vision paradigms.

## Introduction

In the classic framework, vision is a feed-forward process that starts with the analysis of basic features such as oriented edges [1–4]. These basic features are pooled along the visual hierarchy to form more complex feature detectors, until neurons respond to objects [5–9]. A strength of modelling visual perception as such a feedforward process is that it breaks down the complexity of vision into mathematically tractable sub-problems. However, it has become clear that this classic framework cannot account for a wide range of experimental results [10–16].

For example, in a vernier discrimination task, two slightly offset vertical bars are presented in the periphery of the visual field (Fig 1A). The task is to determine whether the bottom bar is offset to the left or to the right. The task is easy when the target is displayed in isolation (Fig 1B, red dashed line). Adding a square around the vernier severely impairs performance (i.e., visual crowding, Fig 1B, first column).

In the classic framework, such impairments are explained by flankers and target features being pooled along the visual hierarchy [17–20]. For example, in Fig 1C, the vernier target and the flankers are pooled, which deteriorates the representation of the vernier. It is often claimed that: a) only elements within the pooling distance, i.e., inside the so-called Bouma's window (equal to half the target eccentricity), affect each other [21–24] and b) adding more flankers within this window always leads to more crowding because more irrelevant information is pooled.

However, recent research has shown many effects that cannot be explained in this classic framework. For example, flankers far from the target (and even far outside Bouma's window) can in fact strongly *improve* performance, depending on the global configuration of the stimulus (uncrowding [10, 11, 25–31]; Fig 1B, second to last columns). As another example, it has been shown that detailed information within Bouma's window can survive crowding [32, 33]. Hence, a) interactions are *not* restricted to Bouma's window and b) adding flankers does *not* always deteriorate information.

Obviously, studies with sparse displays cannot reveal these important effects. Displays that contain a large number of flankers come with the problem that the number of configurations increases exponentially with the number of flankers. For example, a relatively simple array of 8 by 8 either vertical or horizontal flankers has more possible configurations than there are seconds since the Big Bang. Hence, it is hard to determine which configurations show interesting effects that are not captured by the classic framework of vision. How can these configurations be discovered?

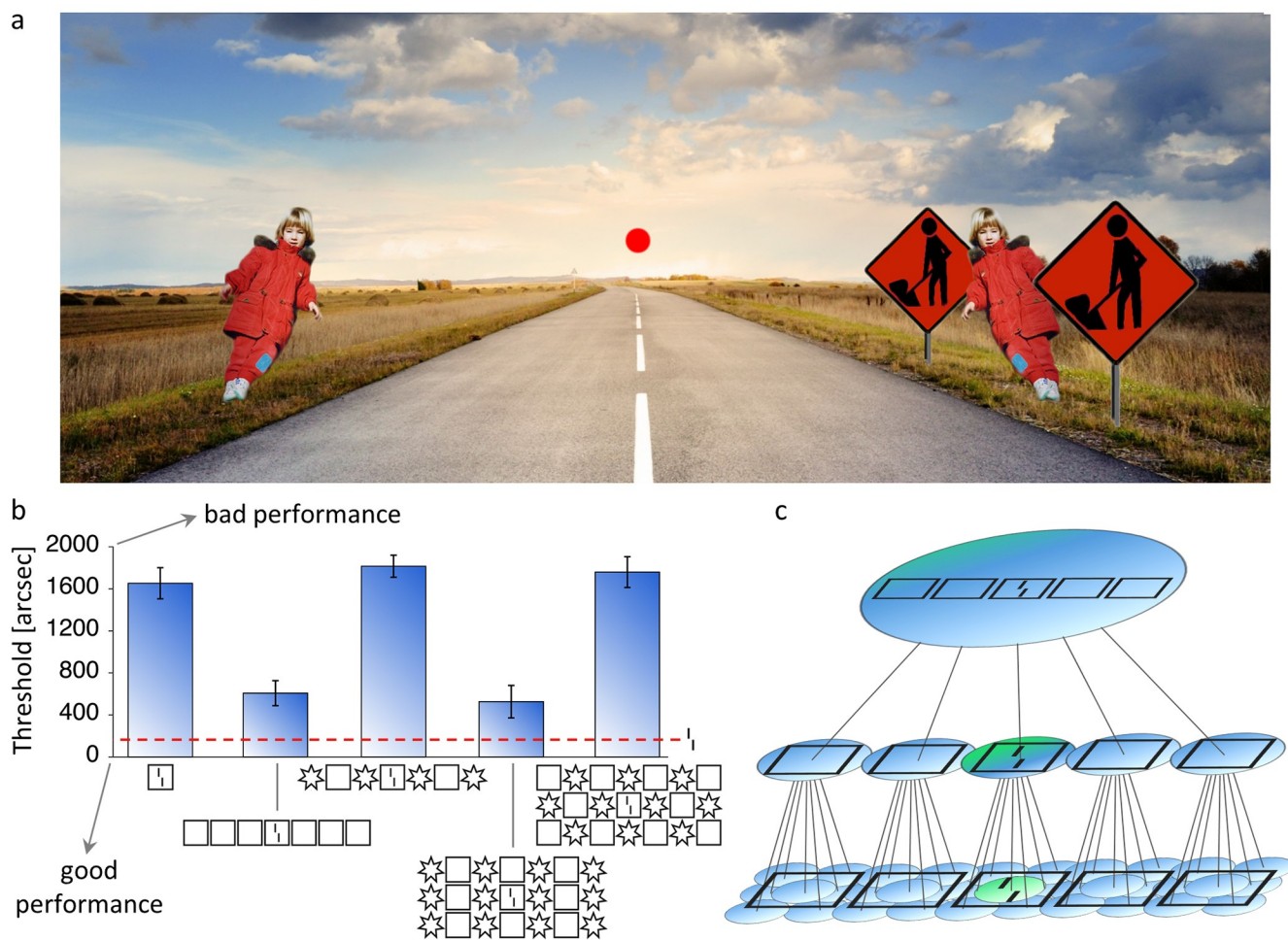

**Fig 1. a. Visual crowding in everyday life.** When looking at the red fixation dot, the child on the right is more difficult to identify than the same child on the left, because the nearby signposts lead to crowding (adapted from [34]). **b.** Manassi et al. [28] presented a vernier in the periphery, surrounded by different flanker configurations. The y-axis shows the vernier offset threshold for 75% of correct responses (the larger the threshold, the worse the performance). In the absence of flankers, the threshold is low (red dashed line). When a square is placed around the target, the task is much harder (crowding, 1st column). When more squares are added, performance recovers almost to the unflanked level (uncrowding, 2nd column). Crowding strength is strongly affected by the whole flanker configuration (3rd to last columns). **c. Classic hierarchical model of crowding.** Local information is pooled along the feedforward hierarchy of the visual system, to form more complex feature detectors. In this example, neurons (circles represent the extent of their receptive fields) detect simple oriented features in the first layer, simple shapes in the second layer and shape configurations in the last layer. Along the hierarchy, pooled activity dilutes information related to vernier offset. In this view, adding more flankers can only lead to stronger crowding. Adapted with permission from [16].

Recently, Van der Burg et al. [35] proposed a paradigm in which observers had to discriminate an almost vertical target, slightly tilted to the left or to the right, embedded in different configurations of vertical and horizontal flankers. First, Bouma's law was verified using *sparse displays*, in which only 4 either vertical or horizontal flankers surrounded the target (Fig 2A). Then, they presented rectangular arrays of 15x19 bars (284 horizontal or vertical flankers and 1 tilted target; *dense displays;* Fig 2B, top). Understanding which distractors at what location interfere with target identification in dense displays is difficult (if not impossible) using a factorial design, as there are $2^{284}$ possible display configurations.

To circumvent this problem, Van der Burg et al. [35] used a genetic algorithm (*GA* [36]; Fig 2B, bottom). In this study, participants performed an orientation discrimination task. For each participant, the displays that led to the highest accuracy were selected and combined using a

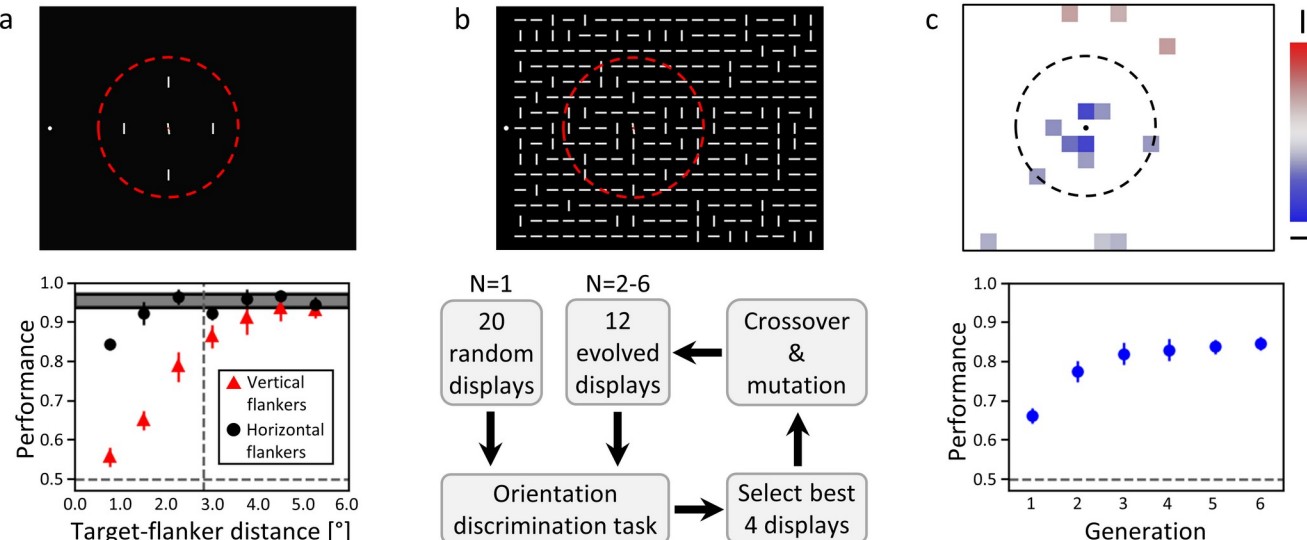

**Fig 2. a. Top.** Example display of the crowding experiment involving sparse displays in Van der Burg et al. [35]. Observers reported whether the target was tilted to the left or right from vertical while fixating the white dot on the left. The target was surrounded by either four horizontal or vertical flankers. The dashed circle, which was not visible during the experiment, indicates Bouma's window. **Bottom.** Human performance (proportion of correct responses) for both flanker orientations and different target-flanker distances. Error bars indicate the standard deviations across observers. The shaded area corresponds to the unflanked condition. The horizontal dashed lines indicate chance level performance. The vertical dashed line indicates Bouma's window. Less crowding was observed for horizontal flankers and Bouma's law was verified. **b. Top.** Example of a dense display. The task was the same as in the sparse display experiment. **Bottom.** GA procedure used in Van der Burg et al. [35]. For every participant, 20 dense displays (whose proportion of vertical flankers was set to lead to 67% of performance) were chosen as the first generation (N = 1). Then the displays that led to the highest accuracy were selected as the parents of the next generation (children). This selection process was repeated for 6 generations of displays (N = 2–6). **c.** Results of the GA procedure in Van der Burg et al. [35]. **Bottom.** During the GA procedure, human performance increased over generations. **Top.** Map depicting which locations in dense displays were crucial for the performance improvements caused by the GA procedure. For each flanker location, the proportion of vertical or horizontal flankers in generation 6, over all participants, was compared (two-tailed t-test) to displays coming from a random selection process between generations (neutral condition). Red/blue slots correspond to locations in which the proportion of horizontal/vertical flankers increased significantly after the evolution process (p < 0.05, not corrected for multiple comparisons to increase the possibility to find evidence for Bouma's law). Colour intensity represents effect size, i.e., in what proportion a vertical or horizontal flanker is selected. White spaces indicate that neither vertical nor horizontal flankers interfered with the target.

crossover and mutation procedure to generate the next generation of displays. This process was repeated over six generations to maximize human performance (see Methods for more details; see [37–39] for a similar methodology to study visual search in complex displays). Using this procedure, performance increased dramatically over generations (Fig 2C, bottom). Interestingly, this improvement was predominantly caused by the target's nearest neighbours and, to a lesser extent, by other flankers within a radius of 1˚ (Fig 2C, top). It seems as if Bouma's window has shrunk.

Here, we investigated which models of crowding can explain these results. To do so, we applied the same GA procedure as in Van der Burg et al. [35], but instead of selecting the displays based on human performance, we selected them based on model performance. First, we tested several leading models of crowding that are based on the classic feedforward pooling framework of vision: a model that artificially reproduced Bouma's law in dense displays (*Bouma model*; see S1 Appendix), a population coding model (*Popcode* model [40]; see S2 Appendix), a model based on summary statistics (*Texture model* [41]; see S3 Appendix) and a feedforward convolutional neural network classifier (*CNN classifier* [16, 42]; see S4 Appendix).

However, we did not expect the former models to reproduce human behaviour for dense displays. Indeed, several studies found that a visual grouping stage is necessary to explain global configuration effects in crowding [15, 16, 43]. For this reason, we also tested several models of crowding that include grouping and segmentation processes: a model of low-level

segmentation (*Laminart model* [44]; see S5 Appendix), a classic convolutional neural network augmented with recurrent grouping processes (*Capsule network* [43, 45]; see S6 Appendix) and a model that combined the population coding and the segmentation models (*Popart model*; see S7 Appendix).

We show that only the models that contain a dedicated grouping mechanism explain the results of Van der Burg et al. [35]. Hence, we propose that grouping is required to explain which elements *within* Bouma's window affect target discrimination performance. Because grouping is also crucial to understand which elements *beyond* Bouma's window impact performance [15], we propose that visual grouping (and not Bouma's law) determines the range of interactions in crowding and naturally and consistently explains why this range highly depends on the nature and the configuration of the visual stimulus.

## Methods

### Ethics statement

Participants gave oral consent before the experiment, which was conducted in accordance with the Declaration of Helsinki except for the preregistration (World Medical Organization, 2013) and was approved by the local ethics committee (Commission d'éthique du Canton de Vaud, protocol number: 164/14, title: Aspects fondamentaux de la reconnaissance des objets protocole général).

The stimuli and the GA procedures were the same as in Van der Burg et al. [35]. We simply replaced human observers with models. The displays were composed of a target (a bar tilted by either +5 or -5 degrees from vertical) embedded in a dense array of 284 flanking bars, each of which was either vertical or horizontal, positioned in a regular and rectangular grid of 15 rows and 19 columns, spanning 11.25˚ by 14.25˚ (see Fig 2B, top, for an example display). Details about how spatial units are represented in each model are given in the appendices. The fixation point (when the tested model used one) was located 0.75˚ to the left of the centre of the left-most column. The target was always displayed at the same position (8th row, 8th column, eccentricity = 6˚) and the task of the models was to report the target orientation (tilted to the left or to the right from vertical). As in the human experiments of Van der Burg et al. [35], model performance for each display was always computed as the proportion of correct responses in 12 trials.

For each model, the GA procedure started with 20 dense displays featuring random configurations of flankers (first generation). The 4 configurations that led to the best model performance were selected as parent configurations. Then, for each model, 12 children configurations were generated by randomly mixing the parent nodes. Each child node had a 50% chance to come from the first parent display and another 50% chance to come from the second one. After this crossover procedure, each node had a 4% chance to be randomly assigned to either a horizontal or a vertical flanker (i.e., a mutation procedure). Those new configurations constituted the next generation of the GA. The same generative process was repeated for 6 generations. To reduce noise, the whole GA was run 4 times, like in Van der Burg et al. [35], where each participant performed 4 sessions.

For each model, we monitored the proportion of vertical and horizontal flankers at each location of the dense displays in the last generation and compared all of them to the respective proportions in the last generation of a random selection process, i.e., a neutral condition, as in Van der Burg et al. [35]. In this neutral condition, the GA parameters were the same as when running the models, except that the displays were selected randomly between the generations. The difference between the model behaviour and the random selection behaviour is presented as a map where a red or a blue slot respectively indicate that the GA procedure selected a

significantly larger fraction of vertical or horizontal flankers at that location, compared to the last generation of randomly selected displays (two-tailed t-tests; $p < 0.05$). Like in Van der Burg et al. [35], the statistical tests were not corrected for multiple comparisons to maximize the possibility of finding evidence for Bouma's law in the results. We call this the *selection measure* (see Fig 2C, top, for corresponding human results). In addition, we made sure that the GA procedure worked, i.e., that model performance increased over generations. We call this the *performance measure* (see Fig 2C, bottom, for corresponding human results). In the results section, we refer to both the performance and the selection measures as the *GA measures*.

In the GA procedure of Van der Burg et al. [35], the proportion of vertical flankers in the first generation of dense displays was set to lead to an initial performance of 67% for each human observer to avoid floor and ceiling effects. Here, we wanted to make a fair comparison between different models. If two models would require for example 10% and 90% of vertical bars, respectively, to have a performance of 67% in the first generation of displays, it would be easier to see a significant increase of horizontal bars in the subsequent generations for the second model than for the first one. For this reason, the initial proportion of vertical flankers was set to a single value for all models, which corresponds to the mean of what was used in Van der Burg et al. [35], i.e., 30% of vertical flankers in the first generation. Prior to the GA procedure, we tuned the parameters of each model to obtain a performance of 67% in dense displays with 30% of vertical flankers. The goal was to find the best parameters for an optimal GA procedure and to have the fairest comparison between models. The performance of some models was bounded by a value lower than 67%. Only in these cases, we adapted the target orientation amplitude so that higher performances than 67% could be reached, thereby allowing the model parameters to be tuned to the required level of performance.

Moreover, we tried as much as possible to tune the model parameters to reproduce Bouma's law in sparse displays. The reason is that our main question was whether the models we tested could reproduce Bouma's law in sparse displays while shrinking their pooling range in dense displays. Hence, we were not interested in models with a small pooling range that could easily reproduce human behaviour in the selection measures, but not Bouma's law in sparse displays. To this end, we measured model performance for the same sparse display experiment as in Van der Burg et al. [35], in which Bouma's law was observed. We call this the *sparse display measure* (see Fig 2A, bottom, for corresponding human results). Finally, we assessed the different models' behaviour for randomly generated dense displays in which the proportion of vertical flankers varied from 0.0 to 1.0 by increments of 0.2. We call this the *proportion measure*. Note that we performed the proportion measure with humans as well, because this experiment was not conducted by Van der Burg et al. [35] (see S8 Appendix for more details).

The GA measures are reported in the Results section by running each model 10 times, to simulate 10 different human subjects, as in the GA procedure of Van der Burg et al. [35]. The reported standard deviations are computed over these 10 runs. The sparse display and proportion measures are not based on statistical testing (i.e., they are not part of the GA procedure) and are hence reported by running each model 100 times, to obtain clearer results. The code for the entire procedure is available at https://bitbucket.org/albornet/shrinking_boumas_window, as well as the code for the different models we tested and instructions to test any other model with the GA procedure.

## Results

Results for all models are summarized in Fig 3. Specific descriptions of the models and details about the results can be found in the supporting information.

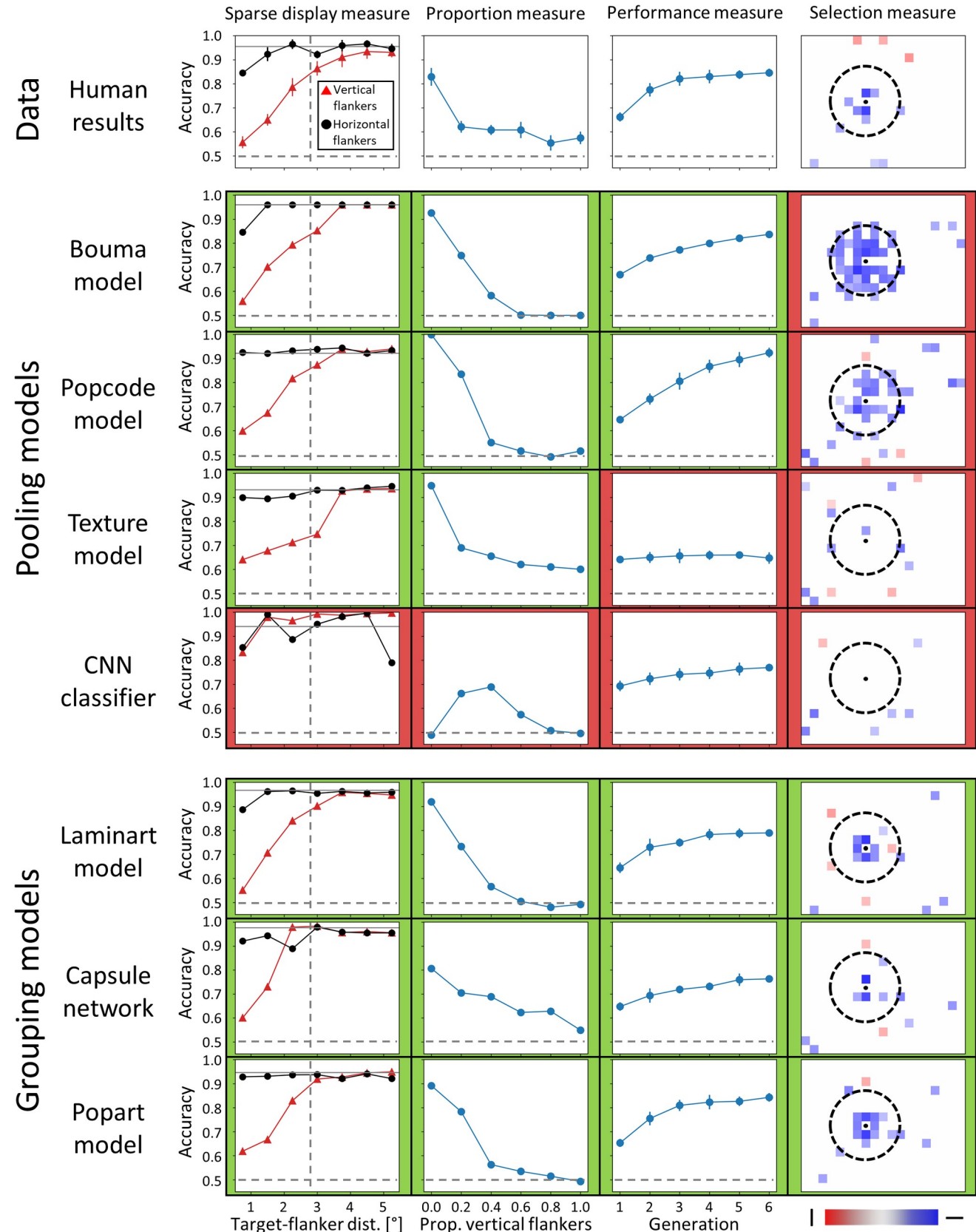

**Fig 3. Results for all models, for the four measures described in the Methods section.** The first row contains the human data. Every column contains a different measure of the models' behaviour and can be compared to the corresponding human data. Each measure is described in detail in the Methods section. For every measure and every model, green/red frames indicate whether a model did/did not qualitatively reproduce the corresponding human data, respectively. For the performance measure, green corresponds to an improvement of at least 10 points of accuracy during the GA procedure. For the other measures, green corresponds to a similar shape in the distribution of the model results and the human data. Note that a quantitative measurement of the similarity between the model results and the human data can be found in S10 Appendix. The vertical dashed lines in the sparse display measure and the dashed circles in the selection measure indicate the limit of Bouma's window. The horizontal dashed lines in all measures indicate chance level accuracy. In general, all models were able to reproduce the sparse display measure and the proportion measure, except for the CNN classifier. Moreover, all models based on the traditional, feed-forward pooling framework of vision failed to reproduce human results for the GA measures (performance and selection measures), either because the GA procedure was unable to find flanker configurations that improved model's performance (Texture model, CNN classifier) or because too many elements within Bouma's window were highlighted by the GA procedure (Bouma model, Popcode model). Finally, all models that contain a grouping stage qualitatively reproduced human results for the GA measures. Note that we included a fine-grained version of the selection measures in S11 Appendix, i.e., in which the fraction of vertical or horizontal flankers selected by the GA procedure is shown for all locations, regardless of whether the differences to the random GA selection process are statistically significant or not.

## Pooling models

First, to rule out the possibility that the GA procedure itself produced the shrinking of Bouma's window, we repeated what was done in Van der Burg et al. [35] and used a simple linear pooling model whose weights were fitted to produce Bouma's law (*Bouma model*). The model qualitatively reproduced the human data for the proportion measure and the sparse display measure but failed to reproduce the human GA measures Fig 3, second row), suggesting that the GA procedure does not produce the shrinking of Bouma's window by itself.

Then, we tested more advanced models based on the traditional, feed-forward pooling framework of vision. First, we used a model based on the population coding idea (*Popcode model* [40]). This model provides a physiologically plausible description of feature integration that accounts for various fundamental features of crowding. Second, we used a model of texture computation (*Texture model* [41]), based on low-level summary statistics, which can be seen as high-dimensional pooling [19]. Texture models may be particularly well suited for dense displays, because they encode complex natural information in a very efficient manner. Third, we used a deep convolutional neural network (*CNN classifier* [42]). Deep neural networks can be seen as a chain of nested pooling and convolution operations. They contain millions of parameters from which unexpected behaviours could arise. The results obtained with these pooling models are shown in Fig 3 (3[rd] to 5[th] rows). Except for the CNN classifier, all pooling models qualitatively reproduced human results for the sparse display and the proportion measures. However, they all failed to reproduce human data for the GA measures, either because no specific configuration was found by the GA procedure to steadily increase model performance (Texture model, CNN classifier) or because too many elements within Bouma's window were highlighted by the GA procedure (Popcode model, Bouma model). More details are in Appendices S1 to S4.

## Grouping models

Finally, we tested several models that describe vision as a two-stage process. In such models, prior to interference such as depicted in the former models, visual elements are parsed into different perceptual groups. Interference only happens after the grouping stage and hence only occurs within these groups. First, we used a model of segmentation based on the recurrent integration of low-level contours (*Laminart model* [44]). The interference stage is the same as in the Bouma model. Second, we used a *Capsule Network*, a type of deep neural convolutional network that includes recurrent processing to implement grouping and segmentation [43, 45]. The results obtained with these models are shown in Fig 3 (6[th] and 7[th] rows). Both models qualitatively reproduced the human results for the sparse display and the proportion measures. Importantly, both models were also able to qualitatively reproduce the human results for the

GA measures: the radius for target-flanker interaction shrank to the nearest neighbour distance.

Despite their success at explaining the shrinking of Bouma's window, these two-stage models face problems of their own. Interference in the Laminart model was fitted to the human sparse measure data, and the Capsule network is difficult to train properly (see S5 and S6 Appendices for details). Exploiting the strengths of visual grouping and of a sophisticated interference mechanism, we combined the Laminart and the population coding models, to test if such a combination would lead to a happy marriage between both families of models (*Popart model*). Indeed, this combined model was able to reproduce human behaviour for all measures (Fig 3, last row; see S7 Appendix for more details).

We also included control simulations in which we tuned the parameters of the pooling models to reproduce human behaviour in the selection measure (instead of the sparse display measure) and checked how they behaved in sparse displays (see S9 Appendix). All pooling models include a parameter that defines their pooling range, which was modified to shrink interactions to the nearest neighbour distance. We show that tuning these models to shrink Bouma's window in dense displays prevents them from reproducing Bouma's law in sparse displays. Hence, only the grouping models can produce a small interaction range in dense displays, while keeping a Bouma-sized range in sparse displays.

## Discussion

To understand crowding and vision in general, paradigms with few elements are the choice to control for complexity and unwanted interactions. For example, based on the traditional framework of vision, many studies have investigated crowding with a target and only a few flankers with a focus on local interference [17–20]. However, such simple paradigms may lead to carved-in-stone principles that are true only in such simple cases but do not apply to realistic situations. As shown here and in many previous publications, this problem seems to manifest in crowding. For example, Bouma's law holds true only for sparse displays [27, 28, 35, 46]. However, complex displays come with their own problems, which are absent in sparse displays. For example, with many flankers, the question is not only *how* visual elements interfere with the target, which is the main question in almost all crowding studies, but also *which* elements interact with each other. In addition, it is difficult to determine which displays to test out of the virtually infinitely many possible ones. To cope with the latter problem, Van der Burg et al. [35] proposed to use a GA procedure to study crowding in dense displays. In their paradigm, among all elements within Bouma's window, only the target's nearest neighbours had an influence on target discrimination performance. Importantly, the shrinking of Bouma's window in dense displays cannot be explained by the large flanker array providing a spatial cue towards the target's location. Indeed, the target is not in the centre of the flanker array. It is at the 8th row and 8th column of a 15 rows by 19 columns flanker array.

Here, we applied the GA procedure to many different models of visual crowding, each coming with its specific hypotheses about the visual system. Such an extensive comparison is a good way to test general principles of vision, because it is possible to identify, among all models, the common causes for the failure or success to explain the results. We have shown that none of the tested models that are based on a cascade of feedforward computations and pooling are able to reproduce the findings of Van der Burg et al. [35]. These models produced results in which either no element or too many elements within Bouma's window were found to interfere. In contrast, all models that include a grouping process could reproduce the human results. It seems that a global grouping and segmentation process is crucial to explain crowding in dense displays. Importantly, the combination of a global grouping stage,

implemented by the Laminart model, and a local interference stage, implemented by the Popcode model, matched human behaviour in sparse as well as in dense displays (Popart model), suggesting that a happy marriage is possible between grouping and pooling models.

Of course, many other models could potentially address these results. For example, we could train a feedforward neural network to only consider the nearest neighbour flankers, since feedforward networks are universal function approximators [47]. However, such a model would be scientifically stale, because crowding is better seen as a probe into visual processing rather than as an explanatory goal per se. A model that only explains this paradigm is useless. For example, feedforward models tuned to explain the shrinking of Bouma's window in the selection measure do not reproduce Bouma's law in sparse displays (see S9 Appendix). The goal is not to overfit on a particular paradigm, but to test how processing characteristics of different hypotheses generalize to this new particular paradigm (crowding in dense displays). CNNs reach human level performance on various complex visual tasks and are subject to crowding. Summary statistics models can explain how humans process complex images without undergoing cognitive overload and capture many characteristics of visual crowding. Segmentation processes are important to solve ill-posed problems of vision and capture the effects of flankers that lie beyond Bouma's window in crowding (e.g. uncrowding). Each of these modelling frameworks has been fruitful in other areas and the goal is to test how they *generalize* to crowding in dense displays, to uncover strengths and weaknesses of each approach.

Along the same lines, Manassi et al. [28] showed that elements beyond Bouma's window can have a strong impact on target discrimination, and that the configuration of elements in the whole visual field determines crowding strength (see also [26, 27]). A similar extensive comparison of models showed, once again, that only models that could reproduce these results contained a dedicated grouping stage [15] (see also [16, 43, 48]). Moreover, Van der Burg et al. [49] showed that crowding in dense displays does not depend on target eccentricity but only on the configuration of the nearest neighbours. The grouping models that we tested here can exhibit uncrowding at the same time as the shrinking of Bouma's window, depending on the specific configuration of the flankers. In contrast, a model that modulates the window of integration based on the number of flankers but not their configuration, such as divisive normalization [50], will not be able to explain why faraway elements interact only in certain cases (e.g. why Manassi and colleagues found very long ranging interactions but Van der Burg and colleagues found the opposite). For all these reasons, it becomes clear that grouping, and not Bouma's window, determines which elements interfere with each other in human vision. In summary, our results do not prove that grouping and segmentation processes are the only way to shrink Bouma's window in dense displays, but rather show that they are the best at explaining crowding overall.

There are many more architectures for feedforward CNNs, such as ResNet and VGG. We think that these networks face similar problems as the feedforward CNNs tested here because they are also based on pooling. Because of this pooling, performance always deteriorates when flankers are added, irrespective of the global configuration of elements (for an in-depth argument, see Doerig et al. [16]). In support of this claim, neither AlexNet (see S4 Appendix) nor the capsule networks controls (see S6 Appendix, feedforward and recurrent CNNs) can explain the shrinking of Bouma's window. In addition, Geirhos et al. [51] showed that CNNs are remarkably consistent with one another behaviourally, irrespective of architecture. In summary, we cannot test all possible models, but have good grounds for proposing that feedforward CNNs cannot explain the flexible range of Bouma's window.

It is important to note that, contrary to our previous work [15, 16], we did not pick the stimuli to pit models against each other. The GA procedure produced the stimuli in a bottom-up fashion. As a limitation for pooling models, we cannot rule out that running the procedure for

more generations may lead to "good" configurations that were not found using only 6 generations. However, there are principled reasons that explain why pooling models do not reproduce human results. Indeed, without grouping and segmentation to "rescue" the target from the flankers, all elements within Bouma's window would decrease performance in those models. Grouping and segmentation seem crucial to explain crowding in general [10, 15, 44, 48]. Moreover, it is known that texture models and other models based on pooling do not reproduce human grouping and segmentation [15, 16, 43, 52, 53]. Hence, it seems unlikely that simply adding generations in the GA algorithm could lead to human-like behaviour. Moreover, even if these models did find interesting configurations after a thousand generations, they would not reproduce an important behaviour, namely, rapid convergence of the GA.

How exactly grouping is implemented in humans is an open question. Here, we have used two different models that include a grouping mechanism. The grouping mechanism in the Laminart model is the formation of illusory contours between well-aligned edges that favour the parsing of visual elements into different layers of the network. This model works particularly well for the kind of displays that are used in Van der Burg et al. [35], because vertical and horizontal elements placed on a regular grid are either perfectly aligned or not aligned at all. However, this mechanism breaks down for more naturalistic stimuli, in which the complexity of low-level edges leads to an excess of illusory contours and, therefore, to bad segmentation. Capsule networks use a fundamentally different mechanism in which grouping is determined by recurrently maximizing the agreement between how neurons interpret a stimulus [45]. This mechanism is much more general than the Laminart model and is a promising candidate as a general framework to understand grouping and segmentation [43]. There are many more possibilities. For example, Linsley et al. [54] proposed another general recurrent grouping mechanism that is scalable to solve complex visual tasks at a state of the art level.

Future research will pit different models of grouping and segmentation against each other. (Un) crowding is one testbed in this respect, but there are many others, for example involving texture segmentation [52, 53], naturalistic image segmentation [54] or spatiotemporal grouping and segmentation [55]. Given the importance of grouping and segmentation, investigating which models can explain these results is an important step towards a better understanding of human vision.

## Supporting information

**S1 Appendix. Bouma's law model.** Detailed description of the model, more details about the results.
(PDF)

**S2 Appendix. Population coding model.** Detailed description of the model.
(PDF)

**S3 Appendix. Texture model.** Detailed description of the model.
(PDF)

**S4 Appendix. CNN classifier.** Detailed description of the model.
(PDF)

**S5 Appendix. Contour segmentation model ("Laminart").** Detailed description of the model.
(PDF)

**S6 Appendix. Capsule network.** Description of the model, including simulation of control models.
(PDF)

**S7 Appendix. Two-stage model ("Popart").** Detailed description of the model.
(PDF)

**S8 Appendix. Human experiment for proportion measure.** Detailed description of the experiment.
(PDF)

**S9 Appendix. Pooling model controls. Control simulations for all pooling models.**
(PDF)

**S10 Appendix. Quantitative similarity measurements.** Assessment of the similarity between the model results and the human data, for Fig 3.
(PDF)

**S11 Appendix. Fine-grained version of the selection measures.** Detailed version of the right-most column of Fig 3.
(PDF)

## Author Contributions

**Conceptualization:** Alban Bornet, Adrien Doerig, Erik Van der Burg.

**Formal analysis:** Alban Bornet, Adrien Doerig.

**Investigation:** Alban Bornet.

**Methodology:** Alban Bornet, Adrien Doerig.

**Software:** Alban Bornet, Adrien Doerig, Erik Van der Burg.

**Supervision:** Michael H. Herzog, Gregory Francis, Erik Van der Burg.

**Visualization:** Alban Bornet.

**Writing – original draft:** Alban Bornet, Adrien Doerig.

**Writing – review & editing:** Alban Bornet, Adrien Doerig, Michael H. Herzog, Gregory Francis, Erik Van der Burg.

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
