## [Decision Letter · Decision Letter 0]

28 Dec 2020

Dear Mr Bornet,

Thank you very much for submitting your manuscript "Shrinking Bouma’s window: Models of crowding in dense displays" for consideration at PLOS Computational Biology.

As with all papers reviewed by the journal, your manuscript was reviewed by members of the editorial board and by several independent reviewers. In light of the reviews (below this email), we would like to invite the resubmission of a significantly-revised version that takes into account the reviewers' comments.

As you will see from the reviewers' comments, there are some technical issues, but mainly there is the concern that the manuscript does not exploit the full potential of the model or that a simpler model might be sufficient to model the data at hand. This is a critical issue also insofar, as for a manuscript deemed to be acceptable to PLoS Computational Biology it is vital to demonstrate that it presents a major step forward beyond the current state of the literature, including earlier work of the same authors. So I urge the authors to carefully consider and address the suggestions made, as even a technically solid manuscript might not be sufficient for PLoS Computational Biology, if it lacks the demonstration of a major conceptual advance.

We cannot make any decision about publication until we have seen the revised manuscript and your response to the reviewers' comments. Your revised manuscript is also likely to be sent to reviewers for further evaluation.

Sincerely,

Wolfgang Einhäuser

Deputy Editor

PLOS Computational Biology

Reviewer's Responses to Questions

**Comments to the Authors:**

Reviewer #1: Bornet et al. investigate how well different models of crowding reproduce the pattern of results from an experiment with human observers. The human observer data stems from an ingenious published experiment, which instead of using simple sparse displays, uses a dense grid of stimuli which is modified over several generations of a genetic algorithm to minimise crowding. In the present manuscript, the authors let the responses of the different models drive the input to the genetic algorithm instead. The key finding is that all four ‘pooling’ models fail to reproduce aspects of the human data, while all three models which include a grouping mechanism produce much closer correspondence.

I read this manuscript with great interest. It clearly written throughout and the conclusions are well-supported by the results of the simulations. I have no serious concerns. I do however have a few suggestions which might improve the manuscript further:

1) Some of the models show substantial variation in the measures depicted in Fig.3. As only 10 ‘participants’ were simulated for each, the results for some of the models are likely to vary quite a bit if the simulation is rerun. As far as I understand, limiting sample size to the human data is necessary to reproduce the ’preference measure’ (rightmost column) as this is based on statistical testing and thus depends on sample size. This is however not the case for the other columns. Would it be possible to base the three leftmost columns on a higher number of simulations to provide clearer data? Perhaps the preference measures derived from those simulations could in addition also be depicted without ‘cutting’ by means of statistical tests to provide more fine-grained information on the range of interference in each model.

2) The different models do not only vary in how well they reproduce the means of the human data, but also in how well they reproduce the standard deviations of the human data (e.g. most of them produce much higher variability in the proportion measure than human observers). The interpretation of the models is however only based on the means, not the standard deviations. I would like the authors to either justify why the models’ ability to reproduce the standard deviations is less important/informative/valid or alternatively, include it in the interpretation of their results.

3) Can the models be evaluated in terms of biological plausibility (the extent to which the computations are compatible with what known about visual cortex)? For example, is it plausible that the computationally more demanding models could be implemented in cortex in such a way as to be compatible with human behavior when stimuli are only presented briefly? Some consideration of this may be worth including in the discussion.

4) The presented simulations yield an argument that grouping may be a sufficient condition for capturing important aspects of human-like behavior in models of crowding. But is it necessary? Put differently, out of the infinity of possible crowding models, are there plausible alternatives to grouping that also yield e.g. a shielding of the target from interference from further away flankers in some conditions (as described in the supplementary material)?

- Lines 187-190: The descriptions here belong in a figure legend rather than the main text

- Lines 203-204: ‘If any model was far from this 67% requirement’: How far is ‘far’?

Reviewer #2: I found this article interesting, in that it uses a display that differs from the one that has repeatedly been used by some of the authors for making similar points. The main message of the article, i.e. the necessity for a grouping process that is not present within typical feedforward architectures, has already been made by some of the authors in previous publications (e.g. Doerig et al 2020 in this same journal), however using uncrowding displays. In this study, the same result is demonstrated using a different class of displays, namely dense arrays of oriented segments that could take on several configurations.

My main issue with this article is that I am not convinced that such a simple set of psychophysical results cannot be explained by a relatively simple model. Unfortunately I don't have time to try and model it myself with code, but the nature of the human results makes me suspect that there should be a way to model them without the necessity for complex algorithms like capsule networks. This is in contrast to uncrowding, a phenomenon for which a relatively low-level strategy seems bound to fail. In other words, while I am more open to the idea that uncrowding may require relatively sophisticated strategies and the grouping idea makes sense, I find it more difficult to convince myself that this is the case for the human data presented here.

As I said, I have no time to model it myself, but I would like to propose some possibilities. For example, suppose we entertain the notion of a flexible attentional window. One model that may provide a bottom-up procedure for shrinking/enlarging the window is the normalization model by Reynolds and Heeger. It may well be that this model makes predictions that are inconsistent with the human data presented here, but perhaps variants of normalization may work. What I have in mind is a situation where the attentional window is relatively large in a sparse display, and then by adding stimulus items, these items stimulate a suppressive attentional field that narrows the focus of attention around the target bar.

Another possibility is the following: suppose my attentional field is driven by a simple heuristic, like "find the centre of the display: that's where the target is". For a sparse display, my estimate of the centre will be very uncertain/inaccurate, so it will be necessary to maintain a relatively large uncertainty/attentional window. For a dense display, where the stimulus items basically trace out the square for me, estimating the centre can be done easily and with high accuracy. A process of this kind can be implemented using low-level algorithms.

There are other possibilities. My point is that I am worried that the authors may not have exhausted all reasonable explanations based on a simpler model. I understand that it is difficult to exclude all such models, as there is potentially an infinite number of them, but I would like to be convinced that one really needs a different class of models to explain this human pattern.

My comments above are also relevant to Figure 3. One suggestion I have with relation to this Figure is that the authors should include a simulation where the "pooling models", or at least some of them, are optimized to replicate the preference measure, so as to demonstrate that this will result in a failure to replicate the other measures. This is important for clarity: some readers may find it puzzling that a simple human pattern like the one in the last column cannot be reproduced, for example, by a CNN. The point is that it cannot be reproduced while at the same time reproducing the sparse-display measure, but I would try and make this message clearer with an explicit simulation, possibly in a separate figure: for example you could have a figure where you present only two simulations for the same pooling model, one simulation for which the model is tailored to the sparse measures, and another one where it is tailored to the dense measures. This figure should demonstrate that you can simulate one or the other, but not both.

I also see issues with the CNN simulations. The authors adopt a slightly unconventional approach, in which they build a read-out module on top of each layer, train that module for a pre-trained Alexnet model, and then choose the layer for which the trained read-out module produces a good match with the sparse measure. I don't have a particular problem with this, but there are many other sensible choices that may produce substantially different results. For example, what happens if you choose the preference (dense) measure as your target and compute cost with respect to that? Will that produce a good match with the sparse measure for free? What if you do this by transfer-learning only the last layer via reducing the fc layer output, as commonly done?

There are many other issues with the section on CNN. Only Alexnet is used. I agree that this is a good example of CNN, but some subsequent architectures cannot be viewed as mere derivatives of Alexnet, and one does not have to look far: even the classic resnet and VGG families have added new features.

It would be reassuring to see more of an effort on the part of the authors to explore these other possibilities (see also my comments immediately above) before jumping directly to their favoured class of models. To some readers, it may appear that the authors were a bit hasty in making the transition. I am not suggesting that this is the case, I am sure the authors have done their job in convincing themselves that pooling models are intrinsically unable to explain the human results, but to a non-committed reader this seems like a difficult conclusion to swallow: after all, the human pattern doesn't look that counter-intuitive. Yes, there is a shrinking window, but that should not be much of a big deal. I think more is needed to convince readers that the failure of pooling models is a necessary conclusion. I should that, from a personal point of view, I am very open to the idea that you need a grouping stage in order to explain this class of phenomena, so I am sympathetic to the conclusions of this paper. But I would still like to be convinced that this is necessary given the data, otherwise this kind of study runs into the risk of simply establishing a "widely accepted notion" that, after a few publications claiming the same, makes people lower their guard.

Minor:

in Supp Material, the authors describe the preference map generated by Popart as being closer to human than Capsule Nets. I did not understand this part. When I look at the human data, I see modulations above and below the target, with little indication of any effect to the sides (left/right). This is what I see for Capsule Nets. Popart, on the other hand, presents a more isotropic pattern, which to me seems less consistent with the human pattern. Please clarify.

Small suggestions:

- in Figure 1b: label y axis with something like "easier/harder" to clarify at a glance that small values are better

- in the Methods section (line 161), stimulus size is described in real spatial units (deg). I understand the authors carried out some additional human measurements for this paper, so it makes sense to specify the stimulus in real-world units, but this paragraph starts by saying that humans were replaced by models. When readers reach the next 3 lines, they'll wonder what it means to define stimuli in deg for a model. So maybe add some clarification here to justify real spatial units.

**Have all data underlying the figures and results presented in the manuscript been provided?**

Reviewer #1: None

Reviewer #2: None

PLOS authors have the option to publish the peer review history of their article (what does this mean?). If published, this will include your full peer review and any attached files.

Reviewer #1: No

Reviewer #2: No
---

## [Decision Letter · Decision Letter 1]

16 Jun 2021

Dear Mr Bornet,

We are pleased to inform you that your manuscript 'Shrinking Bouma’s window: How to model crowding in dense displays' has been provisionally accepted for publication in PLOS Computational Biology.

Best regards,

Wolfgang Einhäuser

Deputy Editor

PLOS Computational Biology

Wolfgang Einhäuser

Deputy Editor

PLOS Computational Biology

Reviewer's Responses to Questions

**Comments to the Authors:**

Reviewer #1: The authors have thoroughly addressed my previous comments. I have no further concerns.

Reviewer #2: I have read the response to my comments and parts of the revised manuscript. I am ok with the revision, however I would like to comment on the fact that, now that the authors have clarified their claim i.e. that some of the existing models can explain these experiments on top of others that have already been demonstrated, this result is undoubtedly of an incremental nature, rather than offering a new concept. At this stage in the submission I do not want to make a big deal about this, so it's ok. But I do want the authors to consider more carefully their future submission to this journal, because at this point their work is starting to look like "salami" science, and this journal is not appropriate for that approach. I like work from this lab, but try and focus on making new points when submitting to PLOS CB, not merely re-iterating the same point over and over again from different angles. The latter approach is also fine for publication, but it should be directed to lower-caliber journals.

**Have the authors made all data and (if applicable) computational code underlying the findings in their manuscript fully available?**

Reviewer #1: Yes

Reviewer #2: Yes

PLOS authors have the option to publish the peer review history of their article (what does this mean?). If published, this will include your full peer review and any attached files.

Reviewer #1: **Yes: **Søren K. Andersen

Reviewer #2: No

---

## [Editor Report · Acceptance letter]

29 Jun 2021

PCOMPBIOL-D-20-01919R1 

Shrinking Bouma’s window: How to model crowding in dense displays

Dear Dr Bornet,

I am pleased to inform you that your manuscript has been formally accepted for publication in PLOS Computational Biology. Your manuscript is now with our production department and you will be notified of the publication date in due course.

With kind regards,

Zsofi Zombor
